# Identification of Multi-Dimensional Relative Poverty and Governance Path at the Village Scale in an Alpine-Gorge Region: A Case Study in Nujiang, China

**DOI:** 10.3390/ijerph20021286

**Published:** 2023-01-10

**Authors:** Zexian Gu, Xiaoqing Zhao, Pei Huang, Junwei Pu, Xinyu Shi, Yungang Li

**Affiliations:** 1Institute of International Rivers & Eco-Security, Yunnan University, Kunming 650500, China; 2School of Earth Sciences, Yunnan University, Kunming 650000, China; 3Forest Resource Management Division, Nujiang Forestry and Grassland Administration, Lushui 673100, China

**Keywords:** multi-dimensional, relative poverty, governance paths, village scale, alpine-gorge region, Nujiang Prefecture, rural regional system

## Abstract

Absolute poverty has historically been solved in China, and the focus on poor areas has shifted to addressing relative poverty. To realize the organic combination of the rural revitalization strategy and relative poverty governance, multi-dimensional relative poverty identification and governance path research at the village scale in an alpine-gorge region is required. For this study, the Nujiang Lisu Autonomous Prefecture’s research area in a typical alpine-gorge was chosen. This paper constructed an evaluation index system for the rural regional system based on location conditions, ecological environment, productive resources, economic base, and public service, based on the theory of multi-dimensional regional poverty and the human–land relationship. The level of poverty, types of poverty, and spatial distribution characteristics of 255 administrative villages were systematically analyzed, and poverty governance paths were proposed. The results show that: (1) There were 215 multi-dimensional relative poverty villages in Nujiang Prefecture, accounting for 84.31% of the total. The relatively poor villages with poverty grades I and II, which are classified as mild poverty, account for 77.21% of all poor villages; this demonstrated that the relatively poor villages in Nujiang Prefecture had a high potential for poverty alleviation. (2) There are 19 different types of constraints in poor villages. Grades III and IV poor villages were mostly found in high-altitude areas. The economic foundation was very weak, the infrastructure was imperfect, the land use type was relatively single, and traffic conditions were relatively backward. (3) The priority model accounted for 16.67% of relative poverty governance, the steady improvement accounted for 28.79%, and key support accounted for 54.54%. Relative poverty governance paths for various counties have been proposed, including rural revitalization priority demonstration, ecological environment governance, eco-tourism, modern agriculture + mountain agroforestry, and improved people’s livelihood and well-being. The findings provided scientific support and direction for future research on the mode and course of relative poverty governance in poor villages in the alpine-gorge area, as well as the rural revitalization strategy’s implementation.

## 1. Introduction

Poverty is a worldwide problem that has afflicted both developed and developing countries [1]. Poverty eradication is the first objective among the 17 Sustainable Development Goals (SDGs) proposed by the United Nations [2]. Home to nearly one-fifth of the world’s population, China completely eradicated extreme poverty—the first target of the UN 2030 Agenda for Sustainable Development—10 years ahead of schedule [3,4,5]. In China, the poor were dispersed across a large geographic area, and there was a complex interaction between poverty and geographic variables. Even after absolute poverty and regional poverty were abolished, it was difficult to eliminate the inherent disadvantages in the geographic location, natural environment, and resource endowments of the places being lifted out of poverty. Rural areas in China face significant differences in economic development; they continue to face issues such as fragile natural environments, lagging infrastructure, and insufficient public services [6,7], and relative poverty will persist [8,9].

Scholars have various interpretations of the concept and connotation of relative poverty; opinions differ on the definition and dimensions of recognizing and evaluating relative poverty. Early research concentrated on the economic dimension [10]. Over time, the connotation of relative poverty has been continuously enriched. In the 1980s and 1990s, Peter Townsend [11] and Amartya Sen [12] proposed the theory of relative deprivation and capability. They emphasized that the dimension of poverty is not limited to material poverty; it also considers a lack of a suitable living environment, rights protection, and other factors that have sparked widespread debate in academia. Ravallion et al. [1] proposed a theoretical model of relative comparisons based on different countries’ relative poverty income lines, calibrating a new data set on national poverty lines. Karahasan and Bilgel [13] analyzed the relative poverty and its geographical distribution in Turkey by developing an evaluation index system that reflected Turkey’s regional development imbalances in education, health care, and living standards. To study relative poverty, an increasing number of scholars have broken through the barrier of income level. The evaluation indicators gradually evolved from a single economic dimension to a multi-dimensional comprehensive welfare analysis of the economy, society, and ecology [14,15,16].

Several studies found it difficult to reduce impoverished people using traditional approaches and ideas when the government helped the poor out of poverty to a specific stage. The remaining impoverished people are concentrated in remote geographical areas with fragile ecological environments, outdated transportation infrastructure, and insufficient public facilities [17,18]. The lack of geographic capital in these regions leads to regional poverty [7], where the government must pull together with more targeted measures and extraordinary efforts. Western Nepal [19] and northern India [20], for example, were primarily mountainous, with a harsh climate, rugged terrain, and slow agricultural industries, which is the most concentrated area of the poor people. Therefore, economic and social factors, as well as geographical environmental factors, such as resource endowment and the fragile environment, should be considered in the study of relative poverty.

The distribution of the rural poor in China is concentrated in 14 contiguous areas of extreme poverty, surrounded by mountains, gorges, karsts, and plateaus [21]. After a victory in poverty alleviation is scored, China comprehensively pushes forward rural vitalization as a historic shift of the focus concerning agriculture, rural areas, and farmers. Rural revitalization and rural relative poverty governance have the same goal of resolving the problem of unbalanced and inadequate development. China’s contiguous areas of extreme poverty are the key rural revitalization areas and the main battlefield of relative poverty governance [7]. Prospective research on relative poverty governance in key rural revitalization areas served as the foundation for achieving sustainable development of the regional economy, society, and environment. Among them, rural poverty in the alpine-gorge environment is characterized by geological disasters, location disadvantages, and backward traffic conditions that distinguish it from poverty in other areas. Promoting the implementation of rural revitalization planning in out-of-poverty areas in the alpine-gorge is a practical problem confronting ecologically fragile impoverished areas, so identifying multi-dimensional relative poverty and proposing governance pathways in these areas is critical.

Over the last decade, methods for identifying multi-dimensional poverty have evolved considerably. Based on field research data, scholars used the Alkire–Foster method [22], multi-dimensional poverty index [23], sustainable livelihood model [24], and other methods to investigate multi-dimensional relative poverty and accurately identify the poor. Scholars used GIS technology to identify relatively poor areas based on night-time lighting data, population income data, and empirical research on the spatial and temporal distribution, evolution law, and driving mechanism of relative poverty to support national and regional poverty reduction decision-making [25,26]. Most current regional studies on relative poverty identification, however, are based on income; scholars defined the relative poverty line by the median proportion of population income, which focused on the economic dimensions and could not reflect the multi-dimensional characteristics of relative poverty [7,27]. Administrative villages are the basic units for implementing China’s rural revitalization strategy and governing relative poverty [28]. The geographical environment and poverty level of each village differ, and different measures should be implemented in different areas. Currently, related research in alpine-gorge areas has used counties as the research unit, but there is a lack of analysis of poverty-causing factors and the spatial distribution of multi-dimensional relative poverty at the village level. A few studies that use administrative villages as evaluation units are mostly small-scale field surveys based on specific sociological research areas [29]. 

The Nujiang Lisu Autonomous Prefecture (Nujiang Prefecture) in a typical alpine-gorge area was chosen as the research area in this paper. The rural regional system’s evaluation index system was built on location conditions, ecological environment, productive resources, economic base, and public service, based on the theory of multi-dimensional regional poverty and human–land relationship. After systematically analyzing the poverty level, poverty types, and spatial distribution characteristics of 255 administrative villages, poverty governance paths were proposed. The research results could provide theoretical support for the government’s local rural revitalization department in carrying out work in accordance with local conditions and promoting the rural revitalization.

## 2. Materials and Methods

### 2.1. Study Area

Nujiang Prefecture is located in the northwest of Yunnan province, China, at 98°39′~99°39′ E, 25°33′~28°23′ N (Figure 1). It connects Diqing, Dali, and Lijiang city to the east, Baoshan city to the south, Myanmar to the west, and Tibet Autonomous Region to the north. The land area of Nujiang Prefecture is 14,700 km^2^ and is divided into four county-level divisions, 29 townships-level divisions, and 255 village-level divisions. The region’s total population was 552,700, with ethnic minorities accounting for 93.60% of the total population, which included 22 nationalities, such as Lisu, Nu, Dulong, Pumi, and others. Nujiang Prefecture is located in northwestern Yunnan’s Hengduan Mountains, which is a landform of “four mountains and three rivers” arranged vertically in sequence (Figure 2). High mountain and canyon landforms cover more than 98% of the area. The land surface pattern of high vertical mountains and deep river valleys arrayed side by side creates obvious regional differences in the region’s natural environment, resource endowment, traffic location, and so on. It has a low-latitude plateau monsoon climate, a distinctive three-dimensional environment, and a forest coverage rate of 78.90%. The geographical setting and climatic conditions have produced an abundance of animal and plant resources. It was not only the core area of the UNESCO World Natural Heritage Site “Three Parallel Rivers of Yunnan Protected Areas,” but it was also the frontier and carrier of the ecological security barrier in southwest China.

Despite Nujiang Prefecture’s abundance of resources, resource utilization was low. Nujiang Prefecture’s ecological environment is fragile, with frequent landslides and mudslides. There is less arable land in the region, and 76.60% of cultivated land has a slope greater than 25°. Mountains and valleys’ topography and landforms create a living environment with inconvenient transportation, information blockade, and long-term closure. It is not only located in China’s concentrated impoverished areas, but it was also one of the deeply impoverished areas in “three districts and three prefectures” of China. In 2020, China’s targeted poverty alleviation policy resolved the problem of absolute poverty in Nujiang Prefecture. However, overall social and economic development remained relatively low. The task of consolidating and expanding poverty alleviation achievements and effectively connecting rural revitalization remains difficult.

### 2.2. Data Source and Processing

In this study, 255 administrative villages were selected as the research object. This study used the Road data, DEM data, NPP data, Meteorological data, Poi data, Geological disaster data, Land cover data, and Economic and social data for the analysis of poverty. Table 1 displays the names, formats, and sources of the data used in this study, the year of data is 2020, except for DEM data. Among them, the Administrative village boundary data and Land cover data were obtained from the Resource and Environment Science and Data Center, which can be used to calculate the arable land, and forest land in the villages. DEM data and Landsat 8 data were obtained from the Geospatial Data Cloud platform, which can calculate the mean slope, elevation, relief intensity, drainage density, and mean NDVI in the villages. The meteorological data were obtained from the China Meteorological Data Network. We used the ArcGIS kriging tool to simulate the overall distribution pattern of annual mean temperature and annual mean precipitation. Road data were obtained from the Open Street Map platform, which can be used to calculate the village road network density and the distance of the village committee from traffic arteries. Geological disaster data were obtained from the Geo Cloud platform, which can be used to calculate the distribution density of geological disasters in the villages. Poi data were obtained from the Gaode map platform, which included educational resources, medical resources, and seats of county government, township government, and village committees. Then, we used the ArcGIS Kernel Density analysis tool to simulate the spatial distribution pattern of educational and medical, and used the ArcGIS Euclidean distance tool to simulate the distance of the village committee from the county government, township government, and resources of educational and medical.

Economic and social data mainly contained the population, cooperative, garden area, labor force population, farmer per capita net income, cultural and natural scenic spots, and so on. The original socio-economic data were collected from field research, statistical yearbooks, and statistical bulletins of counties and cities. In addition, all geographic data were processed by geometric correction, registration, and projection transformation. We extracted data based on administrative village boundaries for a unified analysis.

### 2.3. Research Method

#### 2.3.1. Indicator System for the Rural Regional System

The term “rural regional” refers to the area where the poor live and produce, which was formed by the interaction of society, economy, resources, and environment, and had a distinct structure, function, and inter-regional connection [27]. The region’s social and economic development is hampered by the fragile ecological environment, remote location conditions, and a lack of production resources in the alpine-gorge areas. Therefore, based on the theory of multi-dimensional regional poverty and human–land relationships, combined with the national strategy requirements of comprehensive rural revitalization, and adhering to the principles of data availability, dynamics, and relevance, this paper developed the rural regional system evaluation index system (Table 2).

The level of communication and connection between the village and other areas is an important factor in evaluating rural multi-dimensional poverty. This paper chose the distance from the village committee to the county government, the distance to the township government, the distance from the main traffic line, and the density of the road network to characterize traffic location conditions. The per capita income, labor force, cooperative participation of villagers, and distribution of cultural and natural attractions in the village reflect the village’s economic development background and potential. These indicators assessed the rural economy’s foundation. 

In rural areas, production resources such as arable land, garden land, forest land, and water and heat resources were important economic sources. Temperature and precipitation changes, for example, can reflect changes in heat and water resources to some extent, which is useful for adjusting agricultural planting layouts. The rural ecological environment is the foundation for rural society’s long-term development. The study area is in a geological hazardous area, the ecological environment is fragile, and frequent natural disasters such as mudslides and landslides have seriously hampered socioeconomic development. 

The spatial accessibility of education and medical resources, including the number and the distance to education and medical resources, reflected the quality of public services. To reduce the dimensional influence, the extreme value standardization method (Formulas (1) and (2)) [30] was used to normalize the original data and control the data within a range of [0, 1]. The analytic hierarchy process and the entropy method [31] were used to estimate the index weight on this basis (Table 2).
(1)$$y_{ij}=\frac{\left(x_{ij}-x_{jmin}\right)}{\left(x_{jmax}-x_{jmin}\right)},$$
(2)$$y_{ij}=\frac{\left(x_{jmax}-x_{ij}\right)}{\left(x_{jmax}-x_{jmin}\right)},$$

In the formula, *y_ij_* is the standardized index value; *x_ij_* is the original value of the *j*-th index of the *i*-th dimension, *x_jmax_* and *x_jmin_* are the maximum and minimum of the corresponding index, respectively.

#### 2.3.2. Relative Poverty Identification Method

1.Evaluation of rural single-dimension development potential

The weighted sum formula was used to calculate the potential development score of each administrative village based on the standardized value of each indicator. The rural regional system’s single-dimensional evaluation result represents the village’s development potential in this dimension. The higher the evaluation score, the greater the potential for development potential. The formula is as follows:(3)$$S_{i}=\sum_{j=1}^{n}x_{ij}\times y_{ij}$$

In the formula, *S_i_* is the rural development potential score of dimensions *i*, *i* = 1, 2, 3, 4, 5 represents the location condition, economic base, productive resources, ecological environment, and public service; *x_ij_* is the value of the *j*-th index of the *i*-th dimension; and *y_ij_* is the weight of the *j*-th index of the *i*-th dimension.

2.Single-dimensional relative poverty identification in villages

Because of the adequacy development potential, villagers were able to pursue a certain standard of living at the village level. In contrast, villagers lose the opportunity to pursue that standard and eventually fall into relative poverty due to a lack of development potential [32,33]. If a village’s score exceeded the criterion, it indicated non-relative poverty in this dimension, and vice versa. The formula is as follows:(4)$$P_{i}=\left\{\begin{array}{c} 1(S_{i}\leq Z_{i})\\ 0(S_{i}>Z_{i}) \end{array}\right.$$

In the formula, *P_i_* is the relative poverty score of the *i*-th dimension; the *P_i_* = 1, indicating that the village is in relative poverty in the *i*-th dimension; and *Z_i_* is the critical value of the deprivation of the *i*-th dimension of the development potential. 

The thresholds for the relative scarcity of the rural regional system’s development potential had not yet been unified. This paper compares relative poverty thresholds proposed by scholars [34,35,36]. The natural breakpoint method was used to divide the score of each dimension into three categories, with the lowest score group serving as a criterion of relative poverty assessment.

3.Multi-dimensional relative poverty identification in villages

The single-dimensional relative poverty scores (*P_i_*) were added together to produce the multi-dimensional relative poverty score (*X*), which represents the village’s relative poverty level. When *X* = 0, the village was classified as non-relatively poverty; when *X* = 1, 2, 3, 4, 5, the poverty Grades were I, II, III, IV, V. The formula is as follows:(5)$$X=\sum_{i=0}^{n}P_{i}$$

## 3. Results

### 3.1. Relative Poverty Analysis in Villages

#### 3.1.1. Single-Dimensional Relative Poverty Analysis in Villages

The development potential scores of the villages’ five dimensions were calculated using Formulas (3) and (4). The natural break point classification method was used to divide the development potential of each dimension into three levels. The lowest score group was used as a relative poverty assessment criterion (Figure 3). In Nujiang Prefecture, the number of villages that were in relatively poverty in terms of location condition, economic base, productive resources, ecological environment, and public service was 38, 85, 93, 110, and 77 (Table 3), respectively. The poor villages in Nujiang Prefecture are primarily caused by the fragile ecological environment and a lack of production resources.

The development potential of rural locational conditions gradually decreased from the county seat outward (Figure 3a). The relatively poor villages were mostly concentrated in Lushui city and Lanping county due to poor development conditions (Table 3). The rural economic base in the south was strong, which it was weak in the north (Figure 3b). The relatively impoverished villages were primarily concentrated in Fugong and Gongshan counties due to poor economic fundamental conditions (Table 3). In the central and southeastern regions, rural productive resources were scarce (Figure 3c). The relatively poor villages were mostly concentrated in Lushui city and Lanping county due to a lack of productive resources for development (Table 3). Ecologically vulnerable areas were primarily found in villages along the Nujiang, Dulong, and Lancang rivers (Figure 3d). The relatively poor villages were mostly concentrated in Fugong county, Gongshan county, north of Lushui city, and western Lanping county due to the fragile ecological environment (Table 3). Rural public service levels gradually declined from the county seat outward (Figure 3e). The relatively poor villages were mostly concentrated in Lushui city, Gongshan, and Lanping counties due to the low level of public services (Table 3).

#### 3.1.2. Analysis of Multi-Dimensional Relative Poverty in Villages

The village multi-dimensional relative poverty identification method yielded the multi-dimensional relative poverty score and poverty grade for 255 villages. In Nujiang Prefecture, 215 villages were found to be relatively poverty, accounting for 84.31% of all villages and distributed across four counties (cities) (Figure 4). Non-relative poverty was discovered in 40 villages, accounting for 15.69% of all villages and primarily distributed in Lushui city and Lanping county. 

In the 215 relative poverty villages, the number of relative poverty grades I, II, III, and IV was 84, 82, 42, and 7, respectively. The relatively poor villages with poverty levels I and II, which fall into the mild poverty level, account for 77.21% of all poor villages; this demonstrated that Nujiang Prefecture’s relatively poor villages had a high potential for poverty alleviation.

### 3.2. The Influence and Type Analysis of Relative Poverty Level

#### 3.2.1. Differences in the Impact of Relative Poverty Levels

There were grade differences in the impact of various indicators among different poverty levels based on the poverty grades of the 215 relatively poverty villages in Nujiang Prefecture and the various factors of rural regional development mentioned above in Table 4.

The four indicators of location conditions revealed that the greater the degree of poverty, the worse the location conditions. In terms of location factors, the grade of poverty in relatively poverty villages is clearly related to major thoroughfares and townships. It has a significant inverse relationship with road network density (Table 4).

Regarding the economic base: Economic indicator values further reflected the relationship between various poverty levels and their economic bases. The current income status had a direct impact on their development potential, and the number of laborers was directly related to the village collective’s ability to develop. Furthermore, increasing village income was aided by the development of eco-tourism in alpine-gorge areas. In terms of economic factors, the poverty level of relatively poor villages is significantly negatively correlated with per capita income, labor availability, and the density of cultural and natural attractions (Table 4).

Regarding the productive resources: Land resources were found to be related to development potential, but their effects differed (Table 4). There was no statistically significant relationship between per capita cultivated land and garden area. The density of the river network and the annual average temperature were both beneficial to improving the quality of arable land resources, and there was a significant inverse relationship between the two. The poverty level of the relatively poor villages, on the other hand, had a significant positive correlation with per capita forest area and the average annual precipitation.

Regarding the ecological environment: The alpine-gorge areas’ topography and fragile ecological background made agricultural production and daily life difficult (Table 4). As rural altitude, topographic relief, and the proportion of cultivated land above 25° have increased, so has the relative poverty of the villages. Small differences in the regional slope and vegetation coverage where the villages were located had no significant effect on the improvement of the villages’ development potential. Geological disaster occurrence points in Nujiang Prefecture are mostly concentrated along the Nujiang river, Dulong river, and Lancang river. Because villages along rivers had better location conditions and economic and social development, the poverty level of relatively poor villages was negatively correlated with the distribution density of geological disasters.

In terms of public services, the “bottom line” of social security, education, and medical services was conducive to improving farmers’ and village collectives’ anti-risk capabilities. Obviously, the higher the level of public services, the lower the poverty grade (Table 4).

#### 3.2.2. Classification of Poverty Types in Relatively Poverty Villages

The villages were divided into four types and 19 combinations (Table 5, Figure 5) based on the constraint dimensions and poverty-causing factor combinations of 215 relative poverty villages—single-dimensional restriction, double-dimensional restriction, three-dimensional restriction, and four-dimensional restriction, without five-dimensional restriction. The most common types of poverty villages are by single-dimension and double-dimensions, accounting for 77.21% of all poor villages. The following main poverty-causing combinations of poverty villages were discovered: R (29) > B-E (28) > E (23) > B/R-E-P (17).

Single-dimensional constraint type. This type of poverty village made up 39.07% of all relative poverty villages. Poverty was divided into five categories: location condition constraints, economic base constraints, productive resource constraints, ecological environment constraints, and public service constraints (Figure 5). The ecological environment and the level of public service were the primary constraints for Lushui city; economic development, production resources, and public services were the primary constraints for Lanping county; and economic development was the primary constraint for Fugong county.Double-dimensional constraint type. This type of poverty village accounted for 38.14% of relative poverty villages, and there were nine combinations of “L-R, L-E, L-P, B-R, B-E, B-P, R-E, R-P, and E-P” (Figure 5). The most impoverishing combinations were “B-E,” “R-E,” and “R-P”, accounting for 34.15%, 17.07%, and 15.85%, respectively. The “B-E” type was mostly found in Fugong and Gongshan counties, the “R-E” type in Lushui city, and the “R-P” type in Lanping county.Three-dimensional constraint type. This type of poverty village accounted for 19.53% of relative poverty villages, and there were seven combinations of “L-B-P, L-R-E, L-R-P, L-E-P, B-R-E, B-E-P, and R-E-P”. Three-dimensional constraint types were primarily distributed in Fugong and Lanping counties (Figure 5). There were 14 poverty villages in Fugong county, with the poverty-causing combinations “B-R-E”; there were 15 poverty villages in Lanping county, with the poverty-causing combinations primarily “L-P” collocation.Four-dimensional constraint type. There were seven of these poverty villages, accounting for 3.26% of relative poverty villages, and there are significant geographical, economic, and social disadvantages (Figure 5). Lanping county had three poverty villages; the poverty type was “L-B-R-P”; the ecological environment was better, and the development potential of production resources was limited. In Fugong county and Gongshan county, there were four poverty villages; the poverty type was “L-B-E-P”; the development potential of production resources was higher, but the ecological environment was relatively fragile.

### 3.3. Relative Poverty Governance Pathway

#### 3.3.1. Classification of Relative Poverty Governance Types

Based on an analysis of relative poverty degree and poverty-causing types, this paper assessed the governance types of each relative poverty village. All dimensions of non-relatively poor villages’ development potential were on the rise, making them priority demonstration types of rural revitalization. When the poverty level was I or II, it indicated that the work of relative poverty governance could be completed by overcoming the single or double factors limiting the poverty alleviation of relatively poor villages, and they were thus classified as steadily increasing types. When the poverty level exceeds III, many factors affect the relatively poor villages, so these villages were classified as receiving key assistance. The following are the governance types of 255 villages in Nujiang Prefecture: priority demonstration (40), steady ascending (166), and key assistance (49) (Table 6).

#### 3.3.2. Relative Poverty Governance Path

According to research and field investigations, the constraints of impoverished villages in different counties in Nujiang Prefecture differ noticeably. This paper focuses on resolving outstanding constraints and adhering to the principle of “green development, adapting measures to local conditions, and making up for shortcomings” in order to consolidate and expand poverty alleviation achievements and lay a solid foundation for implementing rural revitalization, and explored the relative poverty governance path (Table 6). Governance paths included: ① Rural revitalization priority demonstration; ② Ecological environment governance; ③ Eco-tourism; ④ Modern agriculture + mountain agroforestry; ⑤ Improved people’s livelihood and well-being. The governance path is as follows:
(1)Rural revitalization priority demonstration mode

The southern part of Lushui city and parts of Lanping county were relatively flat, situated on the main line of external traffic, and served as the focal point of local economic and social development. It benefited from obvious geographical advantages, favorable environmental conditions, and high agricultural output. The area meets the criteria for establishing a priority demonstration area for rural revitalization. Combined with the Chinese government’s “Hundred, Thousand, and Ten Thousand” rural revitalization demonstration project, applied for the construction of rural complexes, boutique demonstration villages, and beautiful villages, could drive the development of neighboring villages.

(2)Ecological environment governance mode

The key to governing relative poverty in Nujiang Prefecture is the co-governance of ecology and poverty. To improve vegetation coverage, it is necessary to strictly enforce the ecological protection red line and the boundary of spatial management and control, as well as to implement ecological restoration projects such as the Grain for Green Project and the governance of slope farmland. The vegetation ecosystem’s soil and water conservation function can be fully utilized to improve the ecosystem’s ability to regulate water and land resources. Geological hazard investigation and prevention must be carried out on a regular basis to increase people’s awareness and ability to prevent and reduce disasters. The governance of the ecological environment requires the participation of both the government and the people. Nujiang Prefecture must actively investigate mechanisms of stable interest connection with villagers, as well as develop rural-based ecological construction forces. Create ecological positions such as ecological forest rangers, natural forest rangers, public welfare forest rangers, and so on through the implementation of ecological industry infrastructure construction. Select and engage local villagers in ecological conservation, as well as promote stable employment for people lifted out of poverty. Consolidate and improve the professional cooperative model for environmental poverty alleviation, and encourage cooperatives to earn labor income by participating in ecological projects such as afforestation, grass planting, and forest tending.

(3)Eco-tourism mode

Although the region’s fragile environment impedes development to some extent, it retains valuable animal and plant, ethnic, cultural, and tourism resources and is one of the world’s biodiversity hotspots [37]. Ecotourism [38,39,40] is a new model of the ecological industry that has high ecological and economic value and may be a suitable choice for green development in mountainous areas, guided by the concept of green development. The canyon’s topography creates a natural scenic spot, which is combined with the Nu, Dulong, Lisu, Bai, and Pumi ethnic groups’ unique national culture. Therefore, developing the ecotourism sector contributes to the protection of the area’s ecological ecosystem while increasing local income.

Local governments must choose a group of towns and villages that have a solid foundation, distinct characteristics, and a strong ethnic cultural heritage, and then build distinctive tourist towns and villages. Increase investment in rural tourism infrastructure and public service facilities, as well as improve the construction of roads, parking lots, restrooms, tourist information services, and other facilities in popular tourist towns and villages. Create a group of rural tourism projects in these distinctive tourist towns and villages, such as agritainments, farmsteads, homestay inns, boutique inns, and leisure agricultural areas. Local villagers are the primary participants in ecotourism, but the majority still lack tourism management skills. The government’s regular rural tourism training has strengthened the training of rural tourism management cadres, village officials, the rural Wealth-Leader, managerial force, agriculture proprietors, and employees.

(4)Modern Agriculture + Mountain agroforestry mode

In the alpine-gorge areas, the prevailing status quo was a scarcity of cultivated and garden land resources and an abundance of forest land resources. Traditional agricultural development has had a limited impact on increasing farmers’ economic income.

Utilizing forest land resources to their full potential and developing mountain agroforestry planting models have high ecological and economic value [41,42]. The plateau gorge mountain region’s geographical environment creates a variety of site conditions and a unique three-dimensional climate that is ideal for growing medicinal plants, edible spices, and edible fungi, as well as planting Fructus Tsaoko, fungi, and Chinese herbal medicines. Under the canopy of forest fruit trees, the high humidity and low light provide an ideal environment for their growth. Fructus Tsaoko, fungi, and Chinese herbal medicines that prefer shade and humidity will thus be planted beneath the forest. Mountain agroforestry mode could improve land resource utilization efficiency, economic income, and environmental environment.

The government encourages farmers to develop moderate-scale characteristic industries based on local conditions through the “enterprise + cooperative + farmers” model, in accordance with the vertical climate of Nujiang Prefecture. Increase farmers’ income by providing more jobs. Xerothermic river valley agriculture should be developed in the Lushui city and western part of Lanping county, this area has abundant heat, abundant light, a large temperature difference between day and night, and good irrigation water conditions, making it suitable for the cultivation of citrus, citron, mango, loquat, pomegranate, and other subtropical fruits. Plateau modern agriculture should be developed in the eastern part of Lanping county. This area’s high altitude, low temperature, and long sunshine hours create ideal climatic conditions for the plateau’s signature cash crops, such as cherries, blueberries, organic vegetables, and honeysuckle flowers.

Develop the green spice industry based on the advantages of regional resources. The Fructus Tsaoko planting area in Nujiang Prefecture was 74,333 hm^2^, and it was an important industry for the villagers to increase their income. Nujiang Prefecture has become China’s primary Fructus Tsaoko production area, as well as the largest Fructus Tsaoko planting area in Yunnan Province. Introduce powerful companies and enterprises to the green spice industry, conduct intensive processing and product research and development focusing on grass fruit, and establish a green spice industry chain system integrating “scientific research + planting + processing + display + sales “ from simple planting to deep processing and brand creation.

(5)Improve people’s livelihood and well-being mode

Continue to improve the “hardware” level of transportation as well as the “software” level of education and health in the alpine-gorge areas. Construction of transportation-based infrastructure could not only drive the development of local social and economic levels but also gradually mitigate the negative effects of location conditions. Simultaneously, continue to promote the implementation of education, health, and other projects to improve people’s livelihoods and well-being, which will contribute to ensuring the bottom line and preventing a return to poverty. Make up for the shortcomings in the village’s road hardening, water supply guarantee, living conditions, and public lighting using the “make up for what is lacking” principle. To create rural communities with thriving businesses, pleasant living environments, social etiquette and civilization, effective governance, and a prosperous plate.

## 4. Discussion

A systematic multi-dimensional indicator system for identifying relative poverty that is consistent with China’s national conditions has yet to be developed. Existing research primarily combines regional characteristics and selects dimensions and associated indicators. Villages with deep regional relative poverty and the primary restrictive factors for alleviating relative poverty have been identified, identifying key focus areas for local government departments to implement rural revitalization strategies and policies [43,44]. For example, Jiao [45] constructed indicators covering three dimensions of health, education, and living standards to assess relative poverty in China’s Qinba Mountains. Chen et al. [46] developed six-dimensional indicators to investigate the relationship between relative poverty and ecological vulnerability in karst landforms. This paper considered the uniqueness of nature and humanities in alpine-gorge areas and constructed an evaluation index system for the rural regional system based on location conditions, ecological environment, production resources, public services, and economic base. It provided a foundation for gradually enriching and improving the multi-dimensional relative poverty identification index system.

In contrast to previous research [47,48,49], as rural distribution density of geological disasters has increased, so has the poverty levels of the villages decreases. However, this did not mean that poverty management in Nujiang Prefecture did not need to pay attention to geological hazards, but rather that geological hazard protection needed to be strengthened. The existing literature focuses on larger regions, such as the entire country or province, and the spatial heterogeneity of geological disasters distribution was evident [50]. Geological disaster occurrence points in Nujiang Prefecture were primarily distributed along rivers and roads [51]. The development of regional water systems, along with road construction, mining and other engineering construction, steep slope farming, and other human activities, created favorable conditions for geological disasters to occur. At the same time, this was the most densely populated, economically prosperous, and industrially developed area in Nujiang Prefecture, with relatively good rural development.

Based on basic village data from 2013, Wang et al. [52] analyzed the poverty-causing factors and types of poor villages in China. It was discovered that five factors were primarily responsible for the poverty-causing types in Nujiang prefecture. The specific factors contributing to poverty include roads, geographical environment, income, social security, cultural quality, and so on. According to this study, single and double dimensions primarily limited relative poverty in Nujiang Prefecture in 2020. The fragile environment and natural resources were the specific causes of poverty. The findings revealed that Nujiang Prefecture had greatly improved in infrastructure, social security, economic income, and other areas during the poverty alleviation period.

This study proposed relative poverty governance pathways suitable for different villages and provided a reference for the sustainable development of rural revitalization in Nujiang Prefecture. However, as for the relative poverty governance pathways of villages in Nujiang Prefecture, only the aspects of industry, eco-tourism, infrastructure, and agriculture were considered in this study. Actually, the suitable poverty governance modes for different villages were various. For example, villages suitable for ecological environment governance mode have significant differences in the workload of ecological environment protection and restoration. As for the villages suitable for eco-tourism mode, it is necessary to determine whether it is suitable for the farmhouse tourism or characteristic tourist attractions, or a mixture of multiple strategies, is more suitable. Additionally, the villages suitable for “modern agriculture + mountain agroforestry” mode have significant differences in the types of favorable agricultural products. Further, the villages suitable for infrastructure are also different. In addition, the time to achieve various indicators varies, and the difficulty of implementation is also different. Therefore, further studies should be carried out to make specific suggestions according to the local regional characteristics and impact mechanism between different measures in the future.

This paper chose the year 2020 as the time node for eradicating absolute poverty in Nujiang Prefecture and began to incorporate relative poverty into the research time scale. Relative poverty research at the village scale can more accurately reflect the background characteristics of rural poverty. However, there are some flaws in this paper that must be addressed. Relative poverty is a dynamic process [53], and a long-term dynamic analysis may better reflect relative poverty changes. In order to develop a dynamic analysis of relative poverty in the future, data will need to be improved further. Regional poverty and individual poverty coexist, with the two having a symbiotic relationship that is interconnected and affects each other [7]; therefore, future analysis of the interaction mechanism relationship should be strengthened. 

## 5. Conclusions

The purpose of this paper was to design a rural regional system for villages in Nujiang Prefecture. It examined the relative poverty level, spatial distribution characteristics, and poverty types in 255 villages using the multi-dimensional relative poverty identification method and proposed the paths of relative poverty governance.
(1)Nujiang Prefecture had 215 multi-dimensional relative poverty villages spread across four counties, accounting for 84.31% of the total. Because of a lack of development potential in the ecological environment and productive resources, the number of poor villages was the highest among them. In the relative poverty villages, the number of relative poverty classes I, II, III, and IV was 84, 82, 42, and 7, respectively. The relatively poor villages with poverty levels I and II, which are classified as mild poverty, account for 77.21% of all poor villages; this demonstrated that the relatively poor villages in Nujiang Prefecture had a high potential for poverty alleviation.(2)In the poverty villages, there were 19 combinations of constraints in the poverty villages; the main impoverishing combinations were R (29) > B-E (28) > E (23) > B/R-E-P (17). When the basic situation of impoverished villages with different poverty levels was compared and analyzed, it was discovered that the relatively impoverished villages with poverty levels I and II had better economic foundations and supporting facilities. They were mostly found near the main road and the town government. The overall facilities and supporting conditions were relatively good, allowing poor households to engage in rural tourism and traditional agricultural industries thanks to certain resource advantages. The relatively poor villages of poverty grades III and IV, in the other hand, had poor natural conditions, with the majority located in high-altitude areas. Most cultivated lands had a slope of more than 25°, the land use type was relatively single, traffic conditions were relatively backward, and the county seat and town government were far away. The economic foundation was very weak, and the infrastructure was inadequate.(3)The 255 villages in Nujiang Prefecture were divided into three types of relative poverty governance: “the priority demonstration” (40, 15.69%), “the steady ascending” (166, 65.10%), and “the key assistance” (49, 19.21%) which requires more attention and assistance. The study adheres to the principle of “green development, adapting measures to local conditions, and making up for shortcomings” pathways to relative poverty governance were proposed for a number of villages, including rural revitalization priority demonstration, ecological environment governance, eco-tourism, modern agriculture + mountain agroforestry, and improved people’s livelihood and well-being. For the development of industries and environmental improvement in poverty-stricken areas, most of the villages were suitable for implementing multi-governing pathways, which highlighted the importance of multidimensional governance. Therefore, it is necessary to pay more attention to the comprehensive integration of multiple modes and form a higher-level comprehensive governance mode by integrating multiple poverty governance pathways in practice to accelerate regional development. This is the focus of the follow-up study.

## Figures and Tables

**Figure 1 ijerph-20-01286-f001:**
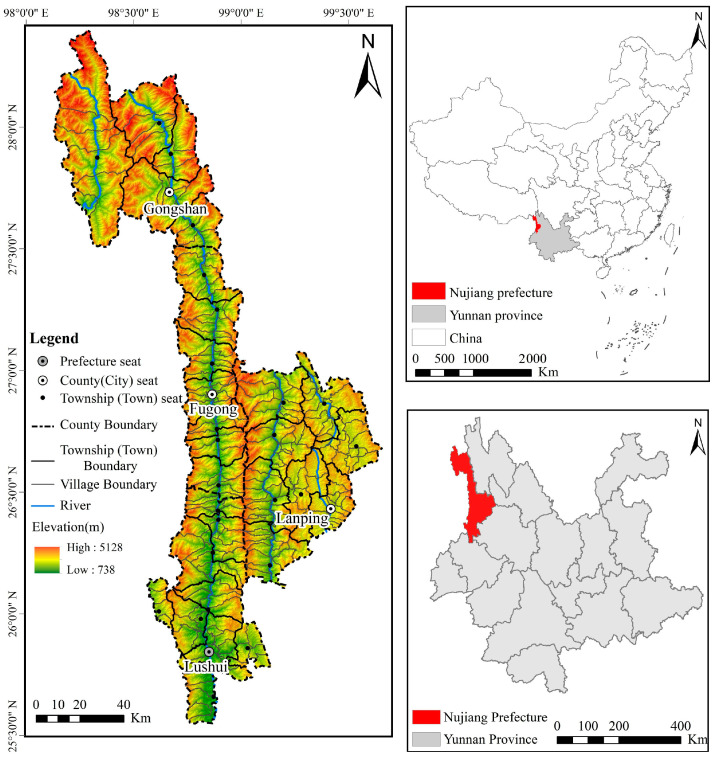
Geographical position of the study area.

**Figure 2 ijerph-20-01286-f002:**
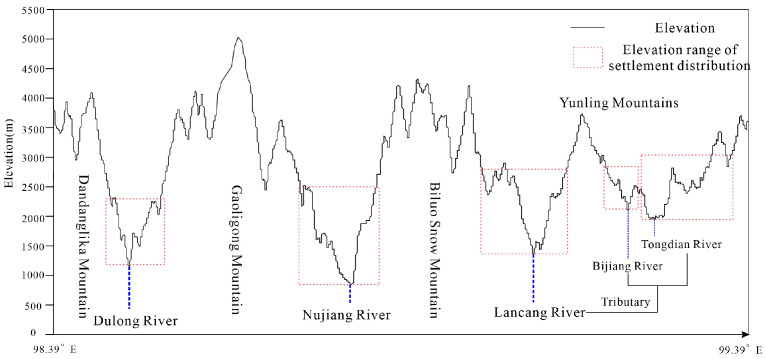
Map of Nujiang Prefecture “four mountains and three rivers”.

**Figure 3 ijerph-20-01286-f003:**
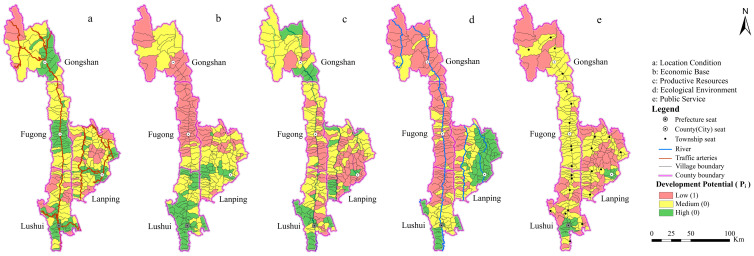
Classification of livelihood capital and relative poverty in villages of Nujiang Prefecture.

**Figure 4 ijerph-20-01286-f004:**
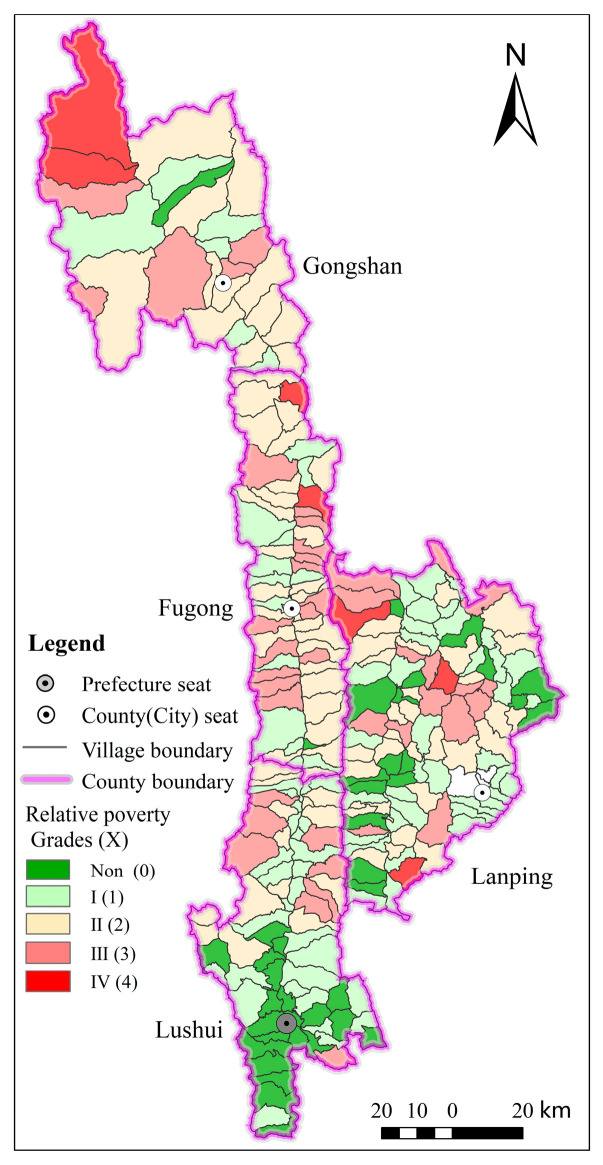
Distribution of multi-dimensional relatively poverty villages in Nujiang Prefecture.

**Figure 5 ijerph-20-01286-f005:**
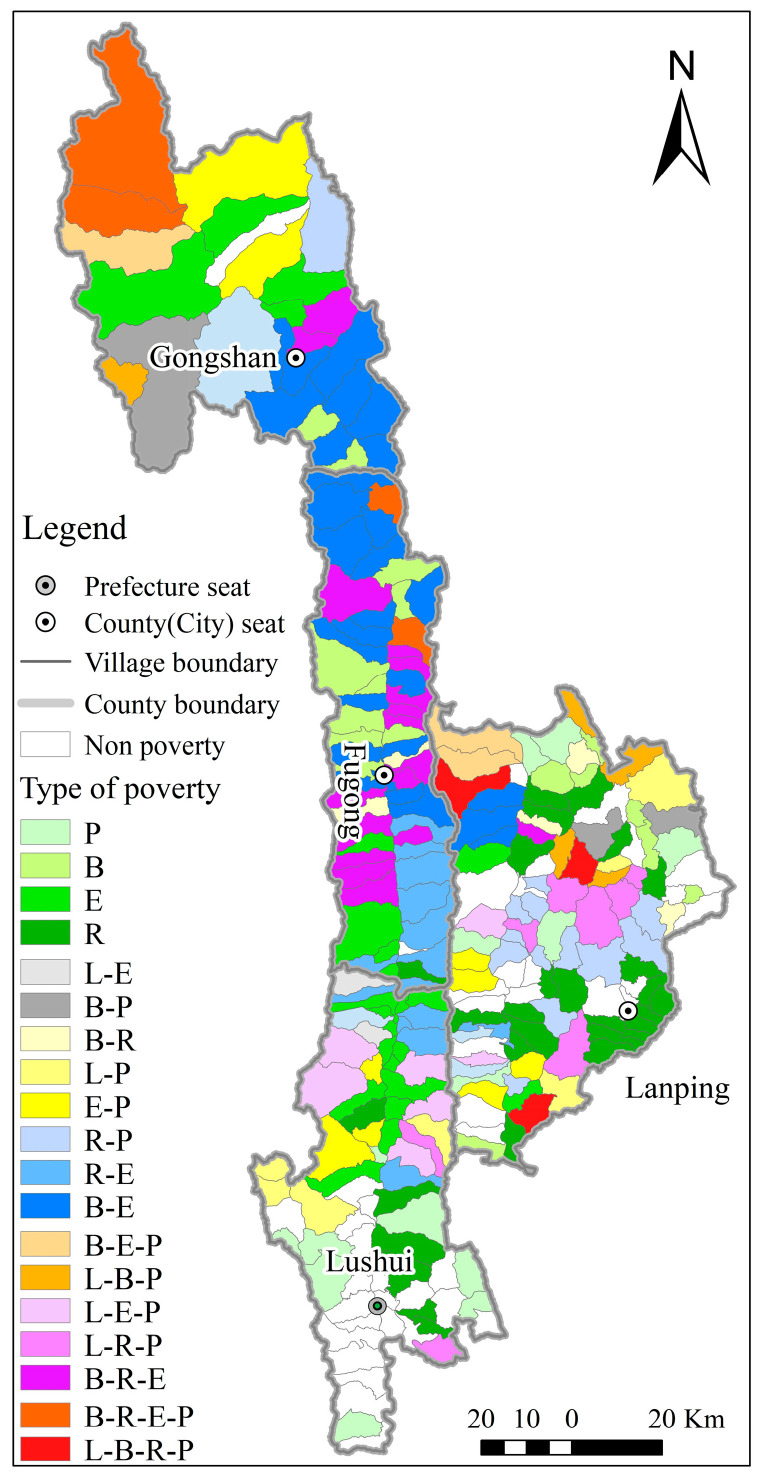
Distribution of relative poverty restriction types in Nujiang Prefecture.

**Table 1 ijerph-20-01286-t001:** Data name, format, and source.

Data Name	Data Format *	Resolution	Data Source
Economic and social data	/	/	The field research, Nujiang Statistical Yearbook and the statistical bulletins of various counties and cities (accessed on 20 February 2022)
Administrative village boundary data	Shp	/	http://www.resdc.cn (accessed on 7 December 2021)
Land cover data	Tiff	30 m	http://www.resdc.cn (accessed on 10 January 2022)
Digital Elevation Model (DEM)	Tiff	30 m	http://www.gscloud.cn (accessed on 10 January 2022)
Landsat 8 data	Tiff	30 m	http://www.gscloud.cn (accessed on 8 February 2022)
Meteorological data	Shp	/	http://data.cma.cn (accessed on 10 February 2022)
Poi data	Shp	/	http://lbs.amap.com/api/webservice/guide/api/search (accessed on 15 February 2022)
Road data	Shp	/	http://www.openstreetmap.org (accessed on 10 February 2022)
Geological disaster data	Shp	/	https://geocloud.cgs.gov.cn (accessed on 17 February 2022)

* Tiff is short for Tagged image file format, Shp is short for Shapefile.

**Table 2 ijerph-20-01286-t002:** Evaluation index system of rural regional system in Nujiang Prefecture.

Dimension	Index	Unit	Index Attribute *	Weight
Location condition	Distance to traffic arteries	km	−	0.3420
	Distance to county	km	−	0.2142
	Distance to township	km	−	0.1882
	Road network density	km/km^2^	+	0.2556
Economic base	Per capita net income	thousand CNY/per	+	0.3794
	Quantity of labor	per	+	0.1927
	Proportion of farmers joining cooperatives	%	+	0.1813
	Density of cultural and natural attractions	number/km^2^	+	0.2466
Productive resources	Arable land area Per capita	hm^2^/per	+	0.2300
	Garden area per capita	hm^2^/per	+	0.1500
	Forest land area per capita	hm^2^/per	+	0.1500
	Drainage density	km/km^2^	+	0.1200
	Average annual temperature	°C	+	0.1500
	Average annual precipitation	mm	+	0.2000
Ecological environment	Slope	°	−	0.1494
	Elevation	m	−	0.1551
	Vegetation coverage (NDVI)	—	−	0.1500
	Relief intensity	m	+	0.1694
	Proportion of area with gradient greater than 25°	%	−	0.1765
	Distribution density of geological disasters	number/km^2^	−	0.1996
Public service	Distribution density of educational resources	number/km^2^	+	0.2150
	Distance to educational resources	km	−	0.3000
	Distribution density of medical resources	number/km^2^	+	0.2250
	Distance to medical resources	km	−	0.2600

* “+” is a positive indicator, and “−” is a negative indicator.

**Table 3 ijerph-20-01286-t003:** Distribution of poverty villages in different watersheds in Nujiang Prefecture.

Development Potential	County (City)				Total
	Lushui	Lanping	Fugong	Gongshan	
Location condition	14 (19.72%) *	20 (19.80%)	1 (1.75%)	3 (11.54%)	38
Economic base	0 (0%)	25 (24.75%)	43 (75.44%)	17 (65.38%)	85
Productive resources	15 (21.13%)	48 (47.52%)	26 (45.61%)	4 (15.38%)	93
Ecological environment	29 (40.85%)	17 (16.83%)	44 (77.19%)	20 (76.92%)	110
Public service	22 (30.99%)	44 (43.56%)	2 (3.51%)	9 (34.62%)	77

* 14 (19.72%): “14” represents that there were 14 poor villages in Lushui city in the dimension of location condition, and “19.72%” represents the proportion of poverty villages in the Lushui city.

**Table 4 ijerph-20-01286-t004:** Indicators of different poverty degrees in relatively poverty villages.

Dimension	Index	Relative Poverty Grades				Pearson Correlation Coefficient
		I	II	III	IV	
Location condition	Distance to traffic arteries (km)	1.19	1.71	3.31	3.73	0.325 **
	Distance to county (km)	28.55	28.17	26.94	36.47	0.015
	Distance to township (km)	5.43	6.55	8.53	11.94	0.334 **
	Road network density (km/km^2^)	0.97	0.65	0.56	0.34	−0.328 **
Economic base	GDP per capita (thousand CNY/per)	10.2	7.80	8.00	6.10	−0.269 **
	Quantity of labor (person)	917	817	754	653	−0.175 *
	The proportion of farmers joining cooperatives (%)	0.09	0.09	0.10	0.09	0.002
	The density of cultural and natural attractions (number/km^2^)	0.0033	0.0032	0.0024	0.0018	−0.202 **
Productive resources	Arable land area Per capita (hm^2^/per)	0.09	0.08	0.08	0.05	−0.049
	Garden area per capita (hm^2^/per)	0.10	0.08	0.07	0.07	−0.095
	Forest land area per capita (hm^2^/per)	2.52	4.59	5.11	15.26	0.263 **
	Drainage density (km/km^2^)	0.74	0.72	0.70	0.70	−0.134 *
	Average annual temperature (°C)	17.69	16.10	15.67	13.27	0.231 **
	Average annual precipitation (mm)	857.78	1019.61	1015.87	1236.63	−0.371 **
Ecological environment	Slope (°)	28.64	30.06	29.89	30.98	0.141 *
	Elevation (m)	1841.95	1802.51	1926.63	2023.14	0.073
	Vegetation coverage	0.64	0.64	0.63	0.57	−0.106
	Relief intensity (m)	46.68	49.30	48.85	50.89	0.129
	The proportion of area with gradient greater than 25° (%)	44	55	57	58	0.196 **
	The distribution density of geological disasters (number/km^2^)	0.0389	0.0379	0.0369	0.0261	−0.079
Public service	The distribution density of educational resources (number/km^2^)	0.72	0.56	0.46	0.27	−0.298 **
	Distance to educational resources (km)	3.37	4.59	6.67	11.68	0.408 **
	The distribution density of medical resources (number/km^2^)	0.78	0.55	0.44	0.26	−0.284 **
	Distance to medical resources (km)	4.99	5.73	8.02	11.92	0.329 **

** and * indicate 1%, and 5% significance levels; otherwise, no significance.

**Table 5 ijerph-20-01286-t005:** Statistics on poverty types in poverty villages.

Grade of Poverty	Main Types of Poverty *				Total
	Lushui	Gongshan	Fugong	Lanping	
I	28 (R/E/P)	13 (B/E)	6 (B/E)	37 (B/R/P)	84
II	14 (R-E)	27 (B-E)	12 (B-E)	29 (R-P)	82
III	8 (L-E-P)	14 (B-R-E)	5 (B-R-E)	15 (L-R-P/L-B-P)	42
IV	0	2 (L-B-E-P)	2 (L-B-E-P)	3 (L-B-R-P)	7

* L, B, R, E, and P represent location condition, economic base, productive resources, ecological environment, and public services.

**Table 6 ijerph-20-01286-t006:** Relative poverty governance pathway.

Types of Governance	Poverty Grades	Governance Pathway				Total
		Lushui	Gongshan	Fugong	Lanping	
priority demonstration	Non	①			①	40
steady ascending	I	②/④/⑤	②/③	②/③	③/④/⑤	84
	II	②-④	②-③	②-③	④-⑤	82
key assistance	III	②-④-⑤	②-③-④	②-③-④	④-⑤/③-⑤	42
	IV		②-③-④-⑤	②-③-④-⑤	③-④-⑤	7

## Data Availability

Not applicable.

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
