# Peer review of "Identification of Multi-Dimensional Relative Poverty and Governance Path at the Village Scale in an Alpine-Gorge Region: A Case Study in Nujiang, China"

_ijerph, 2023, doi:10.3390/ijerph20021286_

Round 1

Reviewer 1 Report

By the submitted manuscript the authors aim to explore the identification method, spatial characteristics, types, and governance path of relative poverty. The topic is important to the formulation of polices for controlling relative poverty, but some problems exist in this manuscript. If the author can seriously modify and perfect it, the manuscript could be accepted after major revision.

Major specific comments:

1.    The language needs to be improved and some errors contain in the article. For example, line 23 the sentence “The result shows that:” repeated; lines 77-78 the sentence “Therefore …. fragile environment.” is incomplete statement.

2.    Line 186 Table1: How are the indicators calculated?

3.    Line 186 Table1, indicators in “Public service”: distribution density and accessibility of educational resources are closely related. Are they duplicated? distribution density and accessibility of medical resources have the same problem.

4.    Line 206, how is the extreme value standardization achieved? Additional clarification is recommended.

5.    Lines 234-236, why did this paper select two critical values? Were the results of the two methods consistent? Critical values of the average score method were not analyzed in section “Results”.

6.    Lines 247-263, what is the data resolution? What is the source of administrative village boundaries?

7.    Lines 273-274, the sentence “The number … productive resources” is not clear.

8.    Section 3.2.1, this section analyzed the relationship between the indicators and relative poverty level. This relationship has been determined in the evaluation of relative poverty level, so it is inevitable that the two should be presented. Moreover, lines 318-349 mentioned many times that the relationship between various indicators and relative poverty level is significant. Has it been statistically tested? The word “significant” cannot be used without statistical test.

9.    Section 3.2.2, The meaning of letters R, L, B, P, E, and S is confusing. What does R represent?

10. Line 491, please check the word “bas”

11. Line 523, please check the sentence “single d double dimensions”

12. Lines 546-548, the sentence “The number … productive resources” is not clear

Reviewer 2 Report

This is a very meaningful research topic, which can improve the consideration of people's life and environment in the region, and it is worth studying. Here's my advice to authors.

1. The conclusion of this paper is very clear, but to achieve the goal, it is necessary to obtain various policy supports from the government. In this regard, it is suggested that the author should make specific suggestions. 2. For the development of industries and environmental improvement in poverty-stricken areas, the time to achieve various indicators varies, and the difficulty of implementation is also different. Therefore, it is suggested that the author should put forward some discussions in "5. Conclusions". as a subject of follow-up research.

Reviewer 3 Report

The paper is not well structured.

Several sentences are inserted in the text without a proper argumentation and with no support by sound references.

The statement “By the end of 2020, China lifted all the rural poor out of poverty by current standards and achieved the poverty reduction target of the UN 2030 Agenda for Sustainable Development 10 years ahead of schedule” is not supported by statistics or dataat national level.

Moreover, several sentences do not have a real and scientific meaning, and are very poor in english language (e.g. "Poverty is a significant challenge facing human development").

Even worse, poor english leads to bad sentences such as the following "Several studies have found that when the poverty-stricken population is eliminated to a particular stage".

As to research method, it creates confusion. In the "Evaluation index system of rural regional system in Nujiang Prefecture", authors put Average annual temperature and Average annual precipitation in productive resources, just to mention one of the several inaccuracies.

Finally, conclusions are scarce and end with a sentence that has no scientific soundness "The study adhered to the principle of "green development, adapting measures to local conditions, and making up for shortcomings" Pathways to relative poverty governance were proposed for various counties, including industrial development, environmental protection, urban-rural integration, infrastructure, and so on".

Round 2

Reviewer 1 Report

The manuscript has been fully improved according to the comments, and I do not have any more comments. The manuscript can be accepted to publication in IJERPH.

Author Response

Dear reviewer,

        Thank you for your careful review of our manuscripts. Your comments are very helpful in improving the quality of our manuscripts. Thank you for your recognition of our work and research.

Reviewer 2 Report

Although the authors have made some explanations, the authors also did his best to make some corrections. As a result, the paper can now be accepted for publication as is.

Author Response

(The authors gave the same response as above.)

Reviewer 3 Report

Despite a series of improvements, overall the paper is still poor especially as to english language (e.g. line 45 "its completes", or the sentence from line 84 to 86, almost ununderstandable).

Methods are poorly outlined (e.g. Table 1).

Punctuation is often wrong (e.g. line 279) and format is poorly respected.

Overall, the paper reflects a low level of attention and scientific soundness.
